

# Synchronous Northern and Southern Hemisphere response of the westerly wind belt to solar forcing

Nathalie Van der Putten[1,3], Florian Adolphi[2], Anette Mellström[3], Jesper Sjolte[3], Cyriel
Verbruggen[4], Jan-Berend W. Stuut[1,5], Tobias Erhardt[2], Yves Frenot[6], Raimund Muscheler[3]

*Correspondence to*: Nathalie Van der Putten (n.n.l.vanderputten@vu.nl)

[1] Department of Earth Sciences, Vrije Universiteit Amsterdam, De Boelelaan 1085, 1081 HV Amsterdam, The Netherlands

[2] Alfred Wegener Institute, Am Handelshafen 12, 27570 Bremerhaven, Germany

[3] Department of Geology, Quaternary Sciences, Lund University, Sölvegatan 12, SE-22362 Lund, Sweden

[4] Department of Geology and Soil Science, Gent University, Krijgslaan 281 (S8), 9000 Gent, Belgium

[5] Royal Netherlands Institute for Sea Research, NIOZ, Texel, The Netherlands

[6] CNRS, EcoBio (Ecosystèmes, biodiversité, évolution) - UMR 6553, University of Rennes 1, Bâtiment 14A, 263 Avenue du Gal Leclerc, 35042Rennes cedex, France

**Abstract.** It has been suggested from observations that the 11-year solar cycle influences regional tropospheric temperature and circulation relatively symmetrically around the equator. During periods of low (high) solar activity, the mid-latitude storm tracks are weakened (strengthened) and shifted towards the equator (poles). The mechanisms behind solar influence on climate are still debated and evidence from paleoclimate records often lacks precise dating required for assessing the global context. Well-dated proxy-based evidence for solar activity and natural climate change exist for the Northern Hemisphere, suggesting pattern similar to today for periods of grand solar minima. However, well-dated and high-resolution terrestrial climate reconstructions are lacking for the Southern Hemisphere. Here we present a unique precisely dated record for past changes in humidity and windiness from the Crozet archipelago at 46° S in the Southern Indian Ocean, a site strongly influenced by the westerly wind belt. We find an increased influence of the westerly winds shortly after 2800 cal year BP synchronous with a major decline in solar activity and significant changes in Northern Hemisphere mid-latitude wind and humidity records. Supported by a general circulation model run encompassing a grand solar minimum, we infer that periods of low solar activity are connected to an equator-ward shift of the mid-latitude westerly wind belts in both hemispheres comparable to the climate reaction to 11-year solar cycle variability inferred from reanalysis data. We conclude that solar forcing is connected to the bipolar climate response about 2800 years ago through synchronous changes in atmospheric circulation of similar sign in both hemispheres.

**Short summary** (maximum 500 characters, spaces included)



In recent decades, Southern Hemisphere westerlies (SHW) moved equator-ward during periods of low solar activity leading to increased winds/precipitation at 46°S, Indian Ocean. We present a terrestrial SHW proxy-record and find

stronger SHW influence at Crozet, shortly after 2.8 ka BP, synchronous with a climate shift in the Northern Hemisphere, attributed to a major decline in solar activity. The bipolar response to solar forcing is supported by a climate model forced by solar irradiance only.

## 1 Introduction

In recent decades, it became clear that the interplay between the latitudinal position of the Southern Hemisphere

Westerly Wind belt (SHW) and the Antarctic Circumpolar Current (ACC), with associated oceanic fronts, plays a crucial role in the Earth's climate system in general, as well as in the role of the Southern Ocean (SO) as a source and/or sink for natural and anthropogenic $CO_2$ (DeVries et al., 2017). Long-term (millennial scale) changes in the intensity and/or position of the SHW affect wind-induced upwelling in the SO and hence, the oceanic meridional overturning circulation and atmospheric $CO_2$ content, during both the last glacial-interglacial transition (Anderson et

al., 2009; Marshall and Speer, 2012; Toggweiler and Samuels, 1995; Toggweiler et al., 2006; Toggweiler, 2009) as the Holocene (Moreno et al., 2010). Superimposed on these long-term changes, centennial scale Southern Annular Mode (SAM) variability related to shifting SHW were found in Southwestern Patagonia, especially during the last 5800 cal yr BP (Moreno et al., 2014; 2018). For the last 3000 years, these warm/dry (SAM+) conditions in Southwestern Patagonia coincide, within dating errors, with warm periods in the Northern Hemisphere (Moreno et al.,

2014). However, the mechanisms behind this interhemispheric symmetry are still unknown, although they probably originate in the atmosphere, for instance through changes in the Hadley cell circulation connected to solar variability (Moreno et al., 2014) . Observations  and reanalysis over the past 50 years reveal a persistent southward shift and strengthening of the SHW during Austral summer, associated with a positive trend of the SAM and contemporaneous with an increase in atmospheric temperatures and $CO_2$ concentrations through upwelling – a potential amplification

of human-induced global warming (Marshall, 2003; Swart and Fyfe, 2012; Thompson et al., 2011).

The discussion and quantitative attribution of natural versus anthropogenic forcing has become a major topic in climate research. Although the topic is still controversial (e.g. Hegerl et al., 2011; Schurer et al., 2013; Ortega et al., 2015) there is increasing evidence for a significant solar influence on climate (Adolphi et al., 2014; Bond et al., 2001; Chambers et al., 2007; Frame and Gray, 2010; Magny, 1993; Moffa-Sanchez et al., 2014; Sjolte et al., 2018; van Geel

and Renssen, 1998). Observational data suggest that global and regional temperatures and atmospheric circulation patterns respond to the 11-year solar cycle, during which total solar irradiance (TSI) varies with about 0.1 % (Gray et al., 2010). Two main mechanisms have been postulated to explain how a relatively small solar forcing can be amplified to produce a significant climate response: (i) the "bottom-up" subtropical coupled air-sea response mechanism (Meehl et al., 2009) and (ii) the "top-down" stratospheric UV ozone mechanism (Haigh,1996). Both mechanisms result in

changing latitudinal temperature gradients and imply a strengthening and broadening of the Hadley circulation and a poleward shift of the mid-latitude storm tracks during years of high solar activity (Meehl et al., 2009, Haigh, 1994).



The complexity of combined natural and human induced climate forcing during recent decades challenges the robust investigations of a solar forced symmetric change in temperature and atmospheric circulation around the equator. However, for the Northern Hemisphere (NH), Sun-climate relationships during periods of grand solar minima such as the Homeric minimum (Stuiver and Kra, 1986), occurring between 2750-2550 cal yr BP (calibrated years before present, AD 1950), have been reconstructed from well-dated proxy-records (Martin-Puertas et al., 2012; Mellström et al., 2015; van Geel et al., 1996). Some recent studies suggest changes in Southern Hemisphere westerly wind belt intensity around 2800 cal yr BP (Heyng et al., 2014; Van der Putten et al., 2008; van Geel et al., 2000) but the chronological control of these proxy records is limited, precluding a reliable investigation of a Sun-climate link in the Southern Hemisphere (SH), and in consequence, impeding a global perspective on climate change during this period.

To address this geographical gap, and add to the Sun-climate debate from a palaeo-perspective we present a unique terrestrial proxy record of wind and precipitation changes from 46°S in the Indian Ocean, a key location with high sensitivity to changes in the SH westerly wind belt and located within the moisture source area of the Antarctic EDC ice core record (Stenni et al., 2001). Key for comparing our SH record with other well-dated records is a high-resolution chronology. A high-resolution "$^{14}$C wiggle-match" age model, based on 15 closely spaced $^{14}$C dates was constructed, with a mean uncertainty of ± 55 years (95.4% probability range) between 2800 and 2500 cal yr BP which is substantially smaller than the duration of the Homeric minimum. Hence, a global perspective on the timing and pattern of climate reaction to a major solar minimum can be assessed for the first time.

## 2 Study site

Ile de la Possession (46°25'S – 51°45 E, Fig. 1a) is a small volcanic island, belonging to the Crozet archipelago situated in the Indian sector of the Southern Ocean (Fig. 1a). The island is characterised by a cool oceanic climate with a mean annual temperature of about 5°C and is strongly influenced by the SH westerly wind belt (Fig. 1a,b). Today, the island is located just north of the core of the westerlies, in the zone of maximal precipitation and strong sea surface temperatures (SST) gradients (Fig 1b). Precipitation on the island is positively correlated to zonal wind speeds just north of the island (Supplementary Fig. S1), making it sensitive to changes in the latitudinal position of the SH westerlies.





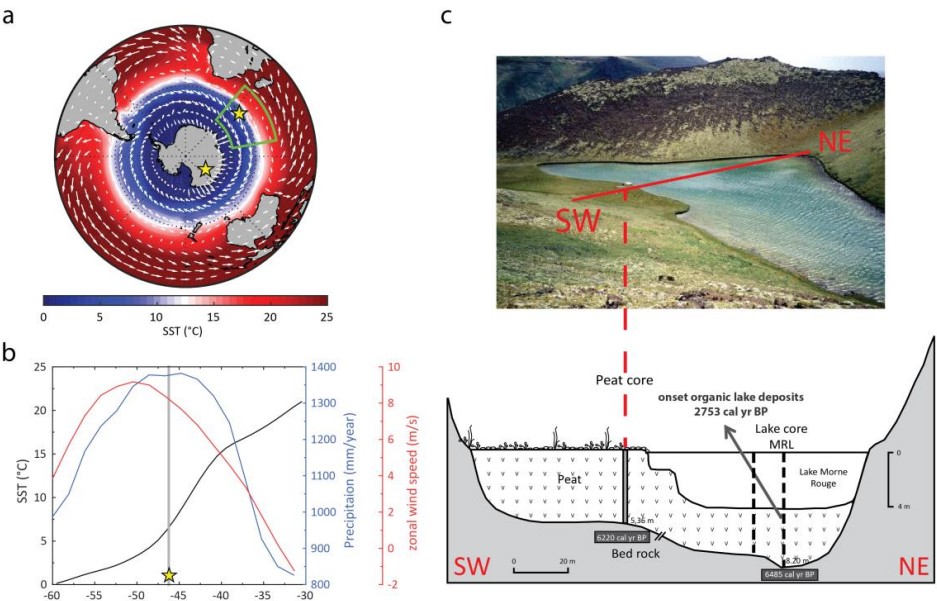

**Figure 1: Crozet archipelago, climatology 1960-2010 CE and study site**. **a,** Annual mean wind speed and direction (white arrows) and sea surface temperatures (colours) from the 20th century reanalysis (Compo et al., 2011). The

yellow stars mark the location of Crozet and EPICA Dome C. **b,** Annual mean zonal mean latitudinal transect of zonal wind speed (red), precipitation (blue) and SST (black) for the green box shown in panel a. The grey line highlights the latitude of Crozet. **c,** Photo of Morne Rouge lake and mire, with a NE-SW cross section of the basin showing the locations of the peat and lake cores with key radiocarbon dates.

**3 Material and methods**

**3.1 Fieldwork**

The Morne Rouge volcanic crater is a closed basin without inlet or outlet (Fig. 1c and Supplementary Fig. S4). A 5.3 m long peat sequence, the Morne Rouge Peat sequence (MRP), covering about 6200 years, was sampled in the Morne Rouge volcano, in a mire bordering a small lake (Fig. 1c) by drilling in two 11 cm diameter PVC tubes (Van der

Putten et al., 2008). A first tube of 2.5 m was hammered to a depth of 235 cm. After retrieving the first tube a second tube of 5 m was put in the same hole and hammered to a depth of 451 cm. Additional coring of the Morne Rouge lake was done from a raft using a 7 cm diameter PVC tube mounted on a modified gauge set. Water depth was about 4 m. This revealed a 4 m thick infilling mostly consisting of peat deposits (Fig. 1c). One of the lake sequences, the Morne Rouge Lake sequence (MRL) was used for further analysis and radiocarbon dating. All cores were transported and

stored at 4°C until further analysis.



### 3.2 High resolution chronology of the Morne Rouge peat sequence

A high-resolution chronology was established based on 15 [14]C dated macrofossil samples around 2800 cal yr BP (Supplementary Information). The samples were pre-treated with NaOH and HCl prior to graphitisation and were measured at the Single Stage AMS facility at Lund University (Adolphi et al., 2013; Skog et al., 2010). All dates are presented in Supplementary Table S1. We applied [14]C wiggle-match dating (Pearson, 1986; van Geel & Mook, 1989) using OxCal version 4.2 (Bronk Ramsey, 2009) with the implemented P_Sequence deposition model (Bronk Ramsey, 2008) and the SHCal13 Southern Hemisphere [14]C calibration curve (Hogg et al., 2013), using a k-value of 0.7. We note that SHCal13 and SHCal20 are virtually identical to each other during our study period, since no new data was added to SHCal20 between 2140 and 3453 cal yr BP (Hogg et al, 2020). The wiggle-matched age-model for the period around 2800 cal yr BP and the subsequent age-depth model for the complete peat sequence are shown in Supplementary Fig. S2 and S3 together with detailed information on the age-model as well as on its robustness.

Additionally, four macrofossil samples of the MRL record were radiocarbon dated (Fig. 1c) in order to determine the onset of highly organic lake deposits (Table S2). Calibration of the single dates was also done in OxCal version 4.2 (Bronk Ramsey, 2009) using the SHCal13 calibration curve (Hogg et al., 2013).

### 3.3 Geophysical and –chemical analysis

In the lab the Morne Rouge peat and lake cores were split longitudinally. Before subsampling, the magnetic low field volume susceptibility (MS) was measured at 2 cm intervals on the MRP, using a Bartington® MS2 susceptibility meter connected to a MS2E1 contact probe having a sensitivity of $\pm1.10-5$ SI unit and a spatial resolution of ~1.5 cm at CEREGE, France (Van der Putten et al., 2008). In the context of this study additional MS measurements at 5 mm resolution for part of the core (185-344 cm), covering the period of the Homeric minimum, were performed using a Bartington Instruments Ltd MS2E1 sensor and a TAMISCAN-TS1 automatic conveyer at Lund University, Sweden (Sandgren and Snowball, 2001).

The latter part of the core was also used for XRF core-scanning. After cleaning and preparation of the archive-halve core surface and covering with SPEXCerti Ultralene® foil, the core was measured with the Avaatech instrument at NIOZ (Texel, the Netherlands) at both 10kV (500μA) and 30kV (150μA) covering the major elements (see supplementary materials for the elements). A step size of 5 mm and a dwell time of 50 sec were used during scanning. The elemental data (raw counts) were normalised to the incoherent + coherent (inc + coh) scattering to account for changes in water content, density and grain size (Davies et al., 2015; Kylander et al., 2012; Van der Putten et al., 2015) before any further (statistical) analysis. Log-ratios of two elements measured by XRF core scanning can be interpreted as the relative concentrations of two elements and minimizes the effects of down-core changes in sample geometry and physical properties (Weltje and Tjallingii, 2008).

To calculate the flux of minerogenic material, the weight of the minerogenic material in a constant volume of peat (mg/cm³) was determined by loss on ignition (LOI) followed by multiplication with the peat accumulation rate (cm/yr) based on the age-depth model.





### 3.4 Plant macrofossil and diatom analyses


Samples (1 cm thickness) for macrofossil analysis were taken every 10 to 15 cm except for the interval between 250 and 310 cm encompassing the Homeric minimum, where high resolution analysis was performed based on the samples taken for the high resolution radiocarbon dating (Supplementary Table S1). Weight and volume of the samples were determined before washing the samples through a 150 µm mesh sieve. Plant macrofossils were quantified in each

sample and are presented as concentrations for the species *Bartramia* cf. *patens* and *Breutelia integrifolia*.

Samples for diatom analysis were prepared following Van der Werff (1955). A total of 500 diatom valves were counted in each slide. The species *Eunotia paludosa* var. *paludosa* and *Stauroforma exiguiformis* are expressed as % of the total sum of diatoms.

For further details on the methods and results for macrofossil and diatom analysis, as well as the full diagrams showing

all taxa, we refer to Van der Putten et al., 2008.

### 3.5 Statistical analysis

Principal Component Analysis (PCA) was applied on the XRF core-scanning data. MS data, reflecting minerogenic material, as well as the ratio of incoherent to coherent (inc/coh) reflecting organic matter (see Davies et al., 2015), were incorporated in the data set for PCA. Data were standardized prior to analysis using the equation $z = (x - \mu)/\sigma$ in

which $\mu$ is the mean and $\sigma$ the standard deviation.

PCA was also applied separately to the plant macrofossil concentration data and diatom counts. Prior to analysis, the values were log-transformed (log (x + 1)).

The PCA Axis 1 (PC1) sample scores of each data set are plotted against age. PCA was performed using the PAST program (Hammer et al., 2001).

To better constrain the onset or start point of the change in our data set, we applied a probabilistic model to the log-transformed MS data (see Erhardt, T. et al., 2019 for more information on the methodology). This method allows the determination of the onset and end of a transition in a data series. It also provides the uncertainties for these points in time resulting from the variability in the data by accounting for this variability in the underlying statistical model.

### 3.6 Reanalysis data

We use the ECMWF twentieth century reanalysis (ERA-20C) (Poli et al., 2016) for analysing the response to solar forcing during the instrumental era (Supplementary Fig. S6). Due to sparse observational data coverage of the southern hemisphere in the early part of the reanalysis we restrict the period for the analysis to 1960-2010.

### 3.7 Modelling





The model simulation is an earth system model sensitivity run performed in connection with the COSMOS Millennium

Experiments (Jungclaus et al., 2008; 2010) model consists of the atmospheric component ECHAM5, surface/vegetation component JSBACH and ocean component MPI-OM/HAMOCC. The run features fixed forcings except for TSI which has a variability (max-min) of 0.25% of the standard TSI value for ECHAM5 (Supplementary Fig. S7).

## 4. Results and discussion

**4.1 Proxy interpretation**

The Morne Rouge peat sequence was taken in a small volcanic crater, without inlet or outlet, in a mire bordering a lake (Fig. 1c). In consequence, changes in hydrology in the peat/lake system are solely depending on atmospheric conditions, such as precipitation, wind intensity and temperature. Coring in the lake revealed that the complete basin was initially filled with peat deposits. Peat initiation started probably shortly after the coming in existence of the

volcano about 6500 cal yr BP (Fig. 1c; Van der Putten et al., 2008). There is no sign of a lacustrine phase in any of the three cores, as could be expected if we would be dealing with a classical hydrosere or terrestrialisation starting with a lake that slowly fills in, ultimately evolving into a mire. Here the lake formed in an existing mire. In the MRL a gradual transition, from peat to a highly organic lake sediment occurs at about 2753 (+95/-207, 2σ) cal yr BP (Table S2).

The peat sequence was analysed for its plant macrofossil and diatom content, complemented by MS and XRF core-scanning measurements, and is supported by a high-precision age model for the interval between 3400-2000 cal yr BP based on 15 [14]C dates (Supplementary Information and Fig. S2, S3, Table S1). The mean calibrated age uncertainty in the age-depth model for all samples is ca. ±65 years (95.4% probability range). For the period from 2800 to 2500 cal yr BP the mean age uncertainties are ca. ±55 years (95.4% probability range). The smallest age uncertainties (ca.

±40 years, 95.4% probability range) are obtained for the samples dated to the two periods of rapid increasing atmospheric [14]C concentration, as reflected in the calibration curve (Fig. S2).

Changes in MS, reflecting minerogenic input as shown by the flux of minerogenic material (Fig. 2a), are interpreted as changes in the wind driven transport of minerogenic particles originating from within the crater as well as from the black sands from the beach situated to the west of the volcano (Supplementary Fig. S4). A change from relatively low

MS values before a transition to relatively high values, with an onset shortly after 2800 years ago, points to increased influx of minerogenic material due to stronger winds (Fig. 2a & c). We quantitatively constrained the onset of this transition from one regime to another, by applying a probalistic model (a fitted ramp) to the log-transformed MS data as shown in Fig. 2b, together with the probability density estimates for the onset (red), midpoint (orange) and end (grey) of the transition. The onset of the transition in the MS data coincides, within 1σ, (68.2% interval 2713-2983,

grey bar in Fig. 2) with the onset of the Homeric solar minimum (black line in Fig. 2).







**Figure 2: Morne Rouge peat sequence and linkage to SH/NH climate and solar forcing**. **a**, Minerogenic influx (red) and magnetic susceptibility (green). **b,** Log-transformed magnetic susceptibility data together with the marginal posterior median of the fitted ramp (black line) with 68.2% (grey line) and 95.4% (grey dashed line) likelihood-intervals. Probability density functions for the onset (red), midpoint (orange) and endpoint (grey) of the fitted transition. The vertical grey bar shows the 68.2% interval for the probability density functions for the onset (red). The black vertical line shows the onset of the Homeric minimum (see j). **c**, First axis (PC1) of a principal component analysis (PCA) on the XRF core-scanning data, including all elements together with MS and inc/coh. **d**, Plant macrofossil content showing the concentrations of the relatively dry species *Bartramia* cf. *patens* (brown bars) and the wet loving species *Breutelia integrifolia* (blue bars). Numbers and "dominant" on the plot refer to concentrations beyond 200. First axis of a PCA analysis including all plant macrofossil taxa (orange). The red rectangle shows the transition with the onset placed at the first occurrence of the wet species *B. integrifolia* and the end at the disappearance of the dry species *B.* cf. *patens*. **e**, Diatom percentages showing transition from the terrestrial/peat species *Eunotia paludosa* var. *Paludosa* (brown) to the lacustrine species *Stauroforma exiguiformis* (blue). First axis of a PCA analyses including all diatom species (orange). Red rectangle as in d. **f**, Deuterium excess (*d*) record of the EPICA Dome C ice core on the AICC2012 chronology (Veres et al., 2013). **g**, Varve thickness of the Meerfelder Maar lake record in Germany ( Martin-Puertas et al., 2012). **h**, Plant macrofossils from a peat bog in the Netherlands supported by high resolution chronology (van Geel et al., 1996 taken from Martin-Puertas et al., 2012) extended with, in light colours, data taken from van Geel (1978) not supported by a high resolution chronology. **i**, Diatom based SST record from a marine record, north of Iceland (Jiang et al., 2015). **j**, Reconstructed total solar irradiance (TSI) based on cosmogenic radionuclides (Steinhilber et al., 2012).

Increased wind driven minerogenic content in our peat sequence, is also obvious from the XRF core-scanning data. We plotted the first axis (PC1) of a PCA performed on a data set, including all elemental data, together with MS and inc/coh. PC1 (explaining 60% of the variance in the data set) is discriminating between organic (inc/coh and Br)¬ and minerogenic content (MS and all other elements) as can be seen in the scatter plot (Fig. S5a). These data support a transition from a period with relatively low minerogenic content to a period with higher minerogenic content at the onset of the Homeric Minimum. PC2 explains only 10 % of the variance and can be regarded as insignificant as the eigenvalue is plotted under the Broken Stick random model curve (Fig. S5b).

Peat archives are excellent climate archives, particularly when they depend exclusively on the atmosphere for their water and nutrient content, such as is the case for ombrotrophic (rain-fed) raised peat bogs (de Jong et al. 2010 and references in there; Li et al., 2020). However, also "geomorphologically isolated" peat systems can be considered as ombrotrophic as is the case for a closed volcanic basin. One way to reconstruct past climate change from peat deposits is to reconstruct changes in bog surface wetness (BSW) which is commonly interpreted as reflecting effective precipitation i.e. the net balance between of precipitation and evapotranspiration and thus as past changes in atmospheric moisture balance (de Jong et al., 2010). Here we use plant macrofossils and diatom analysis as proxies



for BSW (see also Van der Putten et al., 2008). In the plant macrofossil record (Fig. 2d), a gradual transition reflecting

a change to higher BSW starts at 2745 (+60/-45, 2σ) cal yr BP with the first occurrence of the wet moss species *Breutelia integrifolia* and was completed by 2577 (+61/-62, 2σ) cal yr BP when the relatively dry moss *Bartramia* cf. *patens* disappears. This gradual transition is also visible in the complete macrofossil data set as shown by the plant macros PC1 axis (Fig. 2d). Within the diatom data a distinction can be made between terrestrial versus lacustrine environments (Fig. 2e). Diatom communities show a gradual change from a terrestrial/peat environment (*Eunotia*

*paludosa* var. *paludosa)* to a mixed terrestrial (bryophylic)-lacustrine (*Stauroforma exiguiformis)* system between 2745 (+60/-45, 2σ) and 2468 (+74/-75, 2σ) cal yr BP. This is consistent with the change from peat to lacustrine deposits in the lake record indicating the formation of the lake (Fig. 1c) at 2753 (+95/-207, 2σ) cal yr BP, bringing lacustrine diatom species onto the peat surface by temporary flooding and/or wind induced water spray. A short delay in the onset and a more gradual change in the biological proxies in comparison with the minerogenic proxy data can be

expected due to a slower response time of ecosystems to changing conditions (Birks et al., 2000).

In summary, the Morne Rouge 5000 yearlong proxy record points to a regime-shift to wetter and windier conditions occurring shortly after 2800 cal yr BP. Additional evidence for a shift in the SH atmospheric circulation pattern can be found in Antarctic ice core data. The timing of the changes in the peat core coincides with the most pronounced minimum in deuterium excess (*d*) in the EDC ice core for the complete Holocene (Masson-Delmotte et al., 2004) (Fig.

2f). Decreasing *d* is commonly interpreted as decreasing SST in the moisture source area (i.e. north of 50°S in the Indian Ocean) of Dome C and/or as a change in atmospheric circulation in the source area (Aemisegger and Sjolte, 2018; Masson-Delmotte et al., 2004, Stenni et al., 2001; 2011). Considering all proxy-evidence, a shift to wetter and windier conditions can be inferred, that can be explained by an intensification of the SH westerly wind belt on Ile de la Possession from 2780 cal yr BP onward coinciding with a decrease in SST in the area and a sudden drop in total

solar irradiance (TSI) (Fig. 2J).

### 4.2 Comparison with Northern Hemisphere records

Contemporaneous to the changes observed in the SH there are numerous NH records showing sun-climate linkages during the Homeric minimum. Here we only focus our discussion on records that are supported by precise dating control to allow for a robust assessment of the temporal linkages between the Northern and Southern Hemispheres.

Those records represent changes in windiness, humidity and temperature. A lake record from Germany at 50°N (Meerfelder Maar) shows a sharp increase in varve thickness at 2760 (+39/-39, 2σ) cal yr BP due to higher diatom productivity/deposition as a result of wind induced mixing of the water column and thus nutrient availability (Martin-Puertas et al., 2012) (Fig. 2g). Based on a wiggle-matched macrofossil record of a rainwater-fed bog at 52°N in the Netherlands, a transition from the relatively drier *Sphagnum* sect. *acutifolia* to the wetter *Sphagnum imbricatum*

suggests more humid and cooler conditions with an onset shortly after 2800 cal yr BP (Fig. 2h) (van Geel et al., 1996). In addition, SST reconstruction in the North Atlantic at 66°N shows significant negative correlation to solar activity changes for the past 4000 years at centennial time scales, including a clear drop in SST coinciding with the Homeric minimum (Jiang et al., 2015) (Fig. 2i). These climate changes in the NH occur synchronously with those found in the SH and are attributed to changes in the dominant mode of atmospheric variability in the north Atlantic, the North



Atlantic Oscillation (NAO). During a period of low solar activity westerly winds shift equator-ward due to a reduced latitudinal pressure gradient, causing stronger winds and enhanced precipitation over central and Southern Europe, resembling a NAO- phase (Martin-Puertas et al., 2012). In response, atmospheric blocking events occur more frequently (Adolphi et al., 2014; Woollings et al., 2008; 2010) resulting in a weakening of the subpolar gyre south of Iceland and a north-ward transport of colder and fresher surface waters through the North Atlantic Current consistent

with the decrease in SST around Iceland found in proxy-data during decadal to centennial scale periods of low solar activity (Jiang et al., 2015; Moffa-Sanchez et al., 2014).

It is striking that both the Morne Rouge peat record (Fig. 2a-e) and the ombrotrophic peat record from The Netherlands (Fig. 2h), are showing a regime shift, i.e. the system does not shift back although the forcing does (Fig. 2j). It seems that a threshold was crossed in these peat systems. One could claim that, for the Sun-climate link to be robust, the

proxy record should follow the forcing TSI record (Fig. 2j), as is the case, at least for the period considered in Martin-Puertas et al. (2012), for the varve thickness record from Meelfelder Maar (Fig. 2g). However, in the case of a peatland (i.e. an ecosystem), once a transition to a new state occurred, the resulting species composition does not necessarily shifts back to its original state when the forcing vanishes (e.g. Randsalu-Wendrup et al., 2016). Moreover, the dynamical response of the atmosphere to a change in forcing depends on the background climate and has been shown

to differ for different boundary conditions (e.g. Dietrich et al. 2013). Both long-term orbital insolation forcing as short-term solar irradiance variability influence Holocene climate and thus our record. The transition seen in both peat records could be the result of a long term (orbital) trend, pushing the system to cross a threshold resulting in a regime shift. It seems unlikely that a NH and SH peat record, both supported by a high resolution chronology, show a shift, within dating errors, at the same time without being externally forced. I order to obtain such a synchronous signal, a

relatively short forcing is required. Hence, orbital forcing might provide the necessary insolation background climate conditions, but the trigger for the shifting to increased westerly influence on the island is likely connected to the onset of one of the largest and longest Holocene solar minimum. The Homeric minimum forms the onset of a period with series of grand solar minima, following a period of a rather quite Sun between about 5000 and 2800 cal years BP (Stienhilber et al., 2009; Wanner et al., 2014).

**4.3 Mechanisms for a bipolar response to a grand solar minimum**

To study possible mechanisms and modern analogues we investigated anomalies in mean sea level pressure (SLP), zonal wind speed and SST for the austral summer (DJF) between solar minima and solar maxima (Fig. 3 and Supplementary Fig. S6, S7). We use (i) the ECMWF ERA-20C reanalysis data (Poli et al., 2016) and (ii) a 1200-year model run with TSI as only variable forcing (Jungclaus, 2008) (Fig. 3). Solar anomalies from ERA-20C mainly reflect

anomalies related to the solar 11-yr cycle whereas the anomalies in the model run are the response to a prolonged solar minimum comparable to the Homeric Minimum (Supplementary Fig. S6, S7).

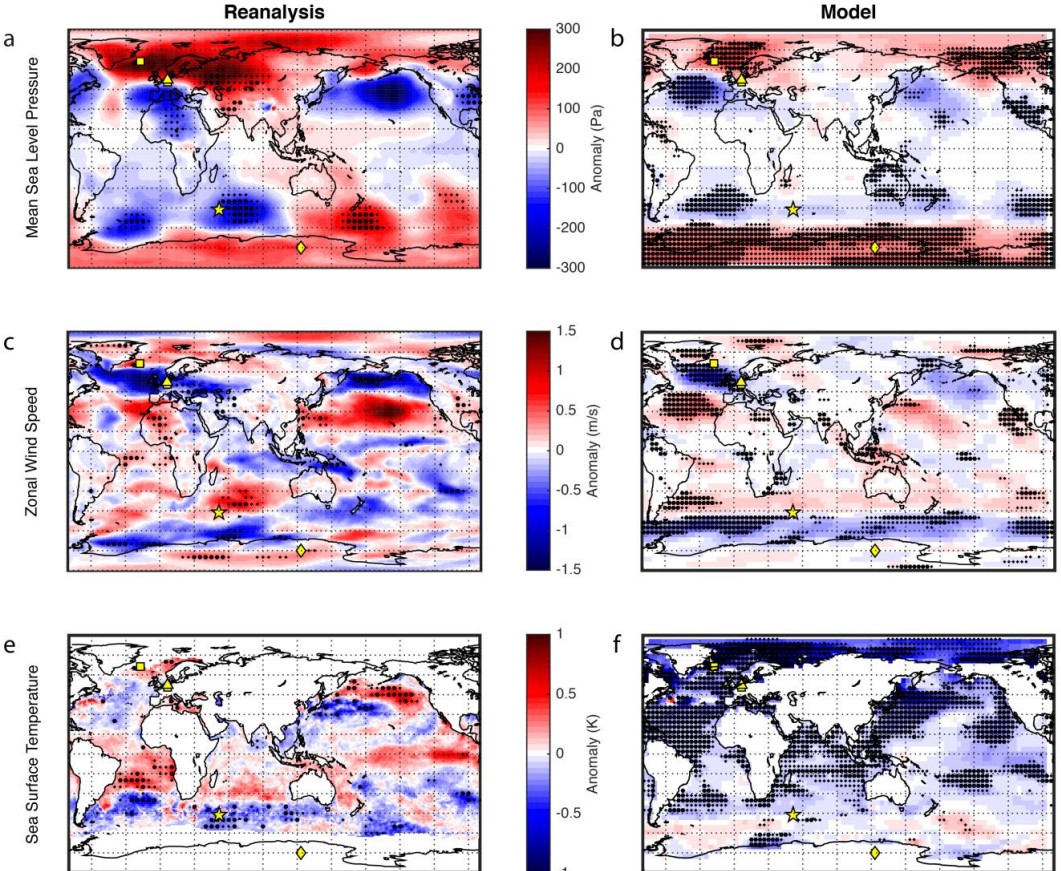

**Figure 3: Solar minimum-maximum anomalies inferred from reanalysis and climate model results. a,b,** mean
sea level pressure, **c,d,** 10 m zonal wind speed and **e,f,** sea surface temperature from the ECMWF ERA-20C reanalysis
(Poli et al., 2016) (**a,c,e**) and a coupled GCM experiment (Jungclaus, 2008; 2010) (**b,d,f**). The solar anomalies have
been defined as shown in Supplementary figures S6 and S7 and indicate anomalies for the solar 11 year cycle (**a,c,e**)
and for Grand Solar Minimum Conditions (**b,d,f**). Large and small black circles highlight significant anomalies at
95% and 90% significance level, respectively, except for the model based sea surface temperature anomalies (**f**) where
they indicate 99% and 95% significance levels. The yellow symbols show the location of paleoclimate records
discussed in the text.

In both reanalysis and model data, reduced solar activity leads to a reduced pressure gradient between the mid- and
high latitudes in both hemispheres resembling a negative phase of the NAO in the North Atlantic region during winter
(Ineson et al., 2011) as well as in the Southern Annular Mode (SAM) in the South Atlantic and Indian Ocean basins
during the austral summer (Fig. 3a,b). In consequence, the westerly wind belts shift equatorward (Thompson and

Wallace, 2000) (Fig. 3c,d) resulting in a strengthening of the westerly influence over the Crozet archipelago at the onset of the Homeric minimum, as indicated by the proxy-data. The spatial pattern of the SST response to solar forcing differs between ERA-20C and the model run. While ERA-20C shows relatively defined zonal bands of SST cooling due to wind changes (Fig 3e) the model response is more globally uniform and related to the global reduction in TSI (Fig. 3f). These two patterns hence reflect the dynamical short term (ERA-20C) and radiative long term (model) SST response to the 11-yr cycle and a grand solar maximum, respectively. However, both datasets indicate a decrease in SST in the area around the Crozet archipelago during periods of low solar activity (Fig. 3e,f) consistent with the deuterium excess (*d*) minimum during the Homeric minimum in the EDC ice core record (Fig. 2f). Although less well-dated our findings in the Indian Ocean are supported by evidence found in a marine proxy record located off the western coast from southern South America suggesting a persistent link between low (high) solar activity and an equator-ward (pole-ward) shift of the SH westerly wind belt during the last 3000 years (Varma et al., 2011; 2012).

**5. Conclusions**

Our results suggest global climate responses to grand solar minima that are relatively symmetric around the equator characterized by an equatorward shift of the westerly wind belts in both hemispheres and a corresponding wettening of the mid-latitudes. These conclusions are qualitatively consistent with modelling scenarios for future grand solar minima (Maycock et al., 2015). They hence provide proxy based support for the hypothesis, that a future grand solar minimum may offset some regional effects of anthropogenic climate change for the duration of the solar minimum.

**Author contributions**

NVdP initiated the study and was responsible for the peat and lake record data and wrote the initial version of the manuscript. CV and NVdP did the fieldwork within the program IPEV 136 led by YF. FA and JS performed the reanalysis and model data analysis. AM prepared [14]C samples and made the wiggle-matched age model. TE applied the probabilistic model, together with the probability density estimates on magnetic susceptibility data. J-BS was responsible for the XRF core-scanning analysis. NVdP, FA, AM, JS, CV and RM jointly discussed the proxy data and climate data. All authors contributed to the discussion and the editing of the final manuscript.

**Competing interests**: The authors declare that they have no conflict of interest.

**Acknowledgements**

This research was made possible thanks to the logistic and financial support of the French Polar Institute (Programme IPEV 136) and the CNRS (Zone atelier de recherches sur l'environnement antarctique et subantarctique). We acknowledge Ryszard Ochyra (Institute of Botany, Polish Academy of Sciences, Krakow, Poland) for helping out with some bryophyte determinations. We thank Bryan Lougheed (Uppsala University, Sweden), Pieter Vroon, Frank Peeters and Sander Veraverbeke (Vrije Universiteit Amsterdam, The Netherlands) for additional analysis and interesting discussion. Rick Hennekam and the late Rineke Gieles (NIOZ, The Netherlands) are thanked for the XRF-analysis. Funding for NVdP, AM, FA and JS was provided by the Swedish Research Council and through a VR





Linnaeus grant to Lund University (LUCCI). A grant to NVdP for radiocarbon dating from LUCCI R3I is greatly acknowledged. RM was supported by the Royal Swedish Academy of Sciences through a grant financed by the Knut and Alice Wallenberg Foundation and the Swedish Research Council (grant DNR2013-8421).

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
