# Peer review of "Synchronous Northern and Southern Hemisphere response of the westerly wind belt to solar forcing"

_Climate of the Past, 2021_

## Author Comment (AC1)

**Final Author Response                                   Van der Putten et al., cp-2021-69**

We take the opportunity offered to us to respond to the review reports of our paper.

Although our paper is rejected for further publication we have to address some of the comments by Referee 1 and 3 that we consider unjustified, as they undermine our scientific integrity without any basis. We believe that a significant number of comments are based on misunderstandings.

Before going into detail we want to summarise our main points:

- Referee 1 reproaches us of "*poor scientific practice*" as we are "*focusing on a single mechanism from the start without evaluating other possibilities*". We agree that in this paper we test the hypothesis of a Sun-climate link, based on chronologically well constrained data together with Reanalysis and Model data. How can testing a hypothesis be considered poor scientific practice? However, we agree that it could be stated more clearly that testing the Sun-climate link is the focus of this paper and that other possible forcing factors could be better discussed.

- Referee 1 and 3 accuse us of "*republishing old data*" and again, "*poor scientific practice*". Indeed, we have revisited a core from an exceptional site at 46°S in the southern Indian Ocean, for which part of the data were published in Van der Putten et al. (2008). However, at that time we could only speculate about solar forcing as our age-depth model was far from accurate enough. For our current paper we obtained a **high resolution chronology**, absolutely necessary for testing a Sun-climate link, together with **additional proxy-data** around the change of interest and supported by **Reanalysis and Model data**, to test our **hypothesis of a solar forced change in the Westerly wind belt**. Interestingly, in the reports of Referees 1 and 3 the words "Reanalysis" and "Model data" are not even mentioned.

- Referee 3 also states that our manuscript "*presents several conceptual and scientific flaws*" which in our opinion are based on misunderstandings by the referee on our proxy-data interpretation as we elucidate below. It seems that Referee 3 thinks that we present the results of a (atmospheric) dust record from a (ombrotrophic) peat bog which is not the case at all for our study as explained below.

Last but not least, we want to highlight that Referee 1 and 2 conclude that our data as well as our interpretations are sound. However, we do agree with Referee 2 that we have to *"provide additional details on our proxy data that will reinforce our interpretations"*. This would probably have avoided certain comments from Referee 3.

**Response to anonymous Referee 1**

We thank Referee 1 (R1) for the comments and suggestions on our manuscript. Below we elaborate on the main points raised by R1.

R1 states that *"our data clearly show a shift from drier to wetter conditions at 2.8 cal kyr BP"* and that *"the interpretation of the data in terms of shift in the southern westerlies is justified"*. Moreover, R1 states that *"the external forcing seems to be justified"* and *"the synchronicity between this record and other records from the NH is a valid argument"*.

We appreciate that R1 acknowledged the quality of our data/record, as well as the interpretation of the data as an intensification of the SHW over the Crozet archipelago, shortly after 2.8 cal kyr BP.

However, R1 (& R3, see below), accuses us of *"poor scientific practice"* as we are *"focusing on a single mechanism from the start without evaluating other possibilities"*.

In this paper we test the hypothesis of a Sun-Climate link, a hypothesis that we cannot reject, based on our chronologically well constrained data combined with Reanalysis and Model data for testing the hypothesised forcing mechanism. How can testing a hypothesis be considered poor scientific practice? We do however agree that it should be more clearly stated in the paper that we are testing the hypothesis of a synchronous equatorward shift in both Hemispheres at the onset of the Homeric minimum.

R1 also reproaches us of "*republishing old data*".

We indeed revisited a core from an exceptional site at 46°S in de south Indian Ocean for which the data are published partially in 2008. However, back then we only could speculate about solar forcing as our age-depth model was far from accurate enough. For our current paper we did a significant amount of additional analyses as we will outline below:
- a high-resolution wiggle-matched [14]C chronology, **absolutely necessary** for testing a Sun-climate link
- a probabilistic model, together with the probability density estimates on magnetic susceptibility data to precisely pinpoint the onset of the change in our data set
- elaborate additional proxy-data around the change of interest
- a comparison with likewise well-dated records from the Northern Hemisphere
- Reanalysis AND Model data to test the hypothesis of a synchronous solar forced change in the westerly wind belts in both hemispheres

So, even if we (partially) reuse published data we believe that there is a wealth of new data to explore further which was only a suggestion in Van der Putten *et al.*, 2008. Interestingly, in the report of R1 the words "high resolution dating", "Reanalysis" and "Model data" are not even mentioned.

We do agree with R1 to be more clear on certain aspects of our paper:
- emphasize what is new in the paper
- add the necessary information and a figure of the lake record
- elaborate the discussion on alternative forcings
- refer to other SH climate records. However, chronology is an issue here as most SH records lack the necessary high-resolution age-depth model.

In response to the following comment of R1 (R3) *"If the mechanism is really global, as the authors argue here, many other mid-latitude SH records should show a shift at 2.8 cal kyr BP"* we would like to argue that this does not have to be the case for several reasons: (i) not all proxies/archives register past climate change in the same way, (ii) as we know, the SHW do not have to act as a hemisphere-wide zonal system, especially when proxy-data from different ocean basins are compared (see fig. 3a, c and e for instance) and (iii) we see no alternative external forcing mechanism that can explain the observed synchronous shifts in both hemispheres (global?) on which we focus in our paper, supporting the suggestion of R1 for *"an alternative explanation is that the SWW shift is long-lasting (multi-centennial/millennial)"*. Also, in our manuscript we did refer to long-lasting orbital forced changes (lines 295-305): *Both long-term orbital insolation forcing as short-term solar irradiance variability influence Holocene climate and thus our record. The transition seen in both peat records could be the result of a long term (orbital) trend, pushing the system to cross a threshold resulting in a*

*regime shift. It seems unlikely that a NH and SH peat record, both supported by a high resolution chronology, show a shift, within dating errors, at the same time without being externally forced. In order to obtain such a synchronous signal, a relatively short forcing is required. Hence, orbital forcing might provide the necessary insolation background climate conditions, but the trigger for the shifting to increased westerly influence on the island is likely connected to the onset of one of the largest and longest Holocene solar minimum. The Homeric minimum forms the onset of a period with series of grand solar minima, following a period of a rather quite Sun between about 5000 and 2800 cal years BP (Stienhilber et al., 2009; Wanner et al., 2015).*

We could have added that, around the period of interest (2800-2700 cal yr BP) there is no evidence from ice cores for exceptional volcanic activity (Cole-Dai *et al.*, 2021; Kobashi *et al.*, 2017; Wanner *et al.*, 2015).

Last, but not least, R1 (R3) also writes *"The interpretation is therefore biased towards solar activity, which uncoincidentally seems to be one of the main research topics of the research group at VU Amsterdam. This is the main issue with this manuscript."*. The subject of the paper is to test the hypothesis of a Sun-climate link and the record presented provides an excellent opportunity to do so. We do not understand how it is relevant that some of the authors worked on the topic before. Also, there is no solar forcing research group at VU Amsterdam. However, we could, as mentioned before, deepen the discussion on alternative forcing mechanisms (but see lines 295-305 in the manuscript).

**References**

Cole-Dai, J., Ferris, D. G., Kennedy, J. A., Sigl, M., McConnell, J. R., Fudge, T. J., Geng, L., Maselli, O. J., Taylor, K. C., and Souney, J. M.: Comprehensive Record of Volcanic Eruptions in the Holocene (11,000 years) From the WAIS Divide, Antarctica Ice Core, Journal of Geophysical Research: Atmospheres, 126, e2020JD032855, 2021.

Kobashi, T., Menviel, L., Jeltsch-Thömmes, A., Vinther, B. M., Box, J. E., Muscheler, R., Nakaegawa, T., Pfister, P. L., Döring, M., Leuenberger, M., Wanner, H., and Ohmura, A.: Volcanic influence on centennial to millennial Holocene Greenland temperature change, Scientific Reports, 7, 1441, 2017.

Steinhilber, F., Abreu, J. A., Beer, J., Brunner, I., Christl, M., Fischer, H., Heikkilä, U., Kubik, P. W., Mann, M., 530 McCracken, K. G., Miller, H., Miyahara, H., Oerter, H., and Wilhelms, F.: 9,400 years of cosmic radiation and solar activity from ice cores and tree rings, Proceedings of the National Academy of Sciences, 109, 5967-5971, 2012.

Van der Putten, N., Hébrard, J.-P., Verbruggen, C., Van de Vijver, B., Disnar, J.-R., Spassov, S., Keravis, D., de Beaulieu, J.-L., De Dapper, M., Hus, J., Thouveny, N., and Frenot, Y.: An integrated palaeoenvironmental investigation of a 6200 year old peat sequence from Île de la Possession, Îles Crozet, sub-Antarctica., Palaeogeography, Palaeoclimatology, Palaeoecology, 270, 179-195, 2008.

Wanner, H., Mercolli, L., Grosjean, M., and Ritz, S. P.: Holocene climate variability and change; a data-based review, 610 Journal of the Geological Society, 172, 254-263, 2015.

---

## Author Comment (AC2)

**Final Author Response**                                    **Van der Putten et al., cp-2021-69**

We take the opportunity offered to us to respond to the review reports of our paper.

Although our paper is rejected for further publication we have to address some of the comments by Referee 1 and 3 that we consider unjustified, as they undermine our scientific integrity without any basis. We believe that a significant number of comments are based on misunderstandings.

Before going into detail we want to summarise our main points:

- Referee 1 reproaches us of "*poor scientific practice*" as we are "*focusing on a single mechanism from the start without evaluating other possibilities*". We agree that in this paper we test the hypothesis of a Sun-climate link, based on chronologically well constrained data together with Reanalysis and Model data. How can testing a hypothesis be considered poor scientific practice? However, we agree that it could be stated more clearly that testing the Sun-climate link is the focus of this paper and that other possible forcing factors could be better discussed.

- Referee 1 and 3 accuse us of "*republishing old data*" and again, "*poor scientific practice*". Indeed, we have revisited a core from an exceptional site at 46°S in the southern Indian Ocean, for which part of the data were published in Van der Putten et al. (2008). However, at that time we could only speculate about solar forcing as our age-depth model was far from accurate enough. For our current paper we obtained a **high resolution chronology**, absolutely necessary for testing a Sun-climate link, together with **additional proxy-data** around the change of interest and supported by **Reanalysis and Model data**, to test our **hypothesis of a solar forced change in the Westerly wind belt**. Interestingly, in the reports of Referees 1 and 3 the words "Reanalysis" and "Model data" are not even mentioned.

- Referee 3 also states that our manuscript "*presents several conceptual and scientific flaws*" which in our opinion are based on misunderstandings by the referee on our proxy-data interpretation as we elucidate below. It seems that Referee 3 thinks that we present the results of a (atmospheric) dust record from a (ombrotrophic) peat bog which is not the case at all for our study as explained below.

Last but not least, we want to highlight that Referee 1 and 2 conclude that our data as well as our interpretations are sound. However, we do agree with Referee 2 that we have to "*provide additional details on our proxy data that will reinforce our interpretations*". This would probably have avoided certain comments from Referee 3.

**Response to anonymous Referee 2**

We acknowledge R2's comments and suggestions. We will change the title to better reflect the content of the paper and we will elaborate on our proxy-methods in more detail, as this is necessary for a wider readership.

R2 shows concerns with the Reanalysis results of our study: "*Section 4.3. Comparison between 11-yr solar cycles in the 'modern" period and the Homeric Minimum falls a bit short of argumentation and description. First, how do these two timescales compare in terms of W/m2 reduction attributable to solar forcing remains unexplored. I fear that comparing 11 year cycles with much longer change in irradiance is a bit like comparing apples and oranges, both in terms of ocean-atmosphere response,*

*and also in terms lag, magnitude and persistence of response in the hemispheric climate as a whole (that includes retroactions with sea ice, thermohaline circulation coupling with pressure fields and wind dymamics etc. ). Also, it is not clear to me how the authors can "'isolate" the spatial patterns of SST variability in the ERA-20C Reanalysis, as solar forcing, if any, is intermingled with many other sources of internal and external variability. Attribution to solar forcing here is risky, and not well supported by analysis. Please improve section 4.3 to document how the attribution (to solar forcing) is made in the ERA-20C which is, contrarily to the 1200 year run, not uniquely forced by TSI."*

We agree with R2 that using the 11-year solar cycle as an analogue to Grand Solar Minima is imperfect and born out of the constraints of the available observations. However, while the oceanic response may be different on longer timescales, the atmospheric (top-down) response does not need to be, since already 11 years is longer than the memory in the atmosphere. More importantly, the lack of a good analogue in the observational record is precisely why we consulted the single-forcing model run. This allows us to overcome the limitations of a short observational period, and demonstrate that the atmospheric response we isolate from the Reanalysis is qualitatively similar to what we obtain for a centennial solar minimum in a state-of-the art climate model that includes the relevant physics. Hence, we believe these two independent results from Reanalysis and Model support each other.

We also agree with R2 that the response to solar forcing is intermingled with all forced and unforced components of the climate system. We want to point out, that our approach to i) exclude years of known abrupt forcing (i.e., volcanoes) and ii) pool the data into composites (or populations) according to the phase of the solar cycle and compare those using classical statistics is an established method (e.g., Ineson *et al.*, 2011; Thieblemont *et al.*, 2015; Woollings *et al.*, 2010). Importantly, this method accounts for the valid concerns of R2: Climate variability, internal or external, not forced by solar variability will contribute to increasing the scatter within each population and thus, lower the significance of a given difference between the populations (Sol. Min vs Sol. Max).

**References**

Ineson, S., Scaife, A. A., Knight, J. R., Manners, J. C., Dunstone, N. J., Gray, L. J., and Haigh, J. D.: Solar forcing of winter climate variability in the Northern Hemisphere, Nature Geoscience, 4, 753-757, 2011.

Thiéblemont, R., Matthes, K., Omrani, N.-E., Kodera, K., and Hansen, F.: Solar forcing synchronizes decadal North Atlantic climate variability, Nature Communications, 6, 8268, 2015.

Van der Putten, N., Hébrard, J.-P., Verbruggen, C., Van de Vijver, B., Disnar, J.-R., Spassov, S., Keravis, D., de Beaulieu, J.-L., De Dapper, M., Hus, J., Thouveny, N., and Frenot, Y.: An integrated palaeoenvironmental investigation of a 6200 year old peat sequence from Île de la Possession, Îles Crozet, sub-Antarctica., Palaeogeography, Palaeoclimatology, Palaeoecology, 270, 179-195, 2008.

Woollings, T., Lockwood, M., Masato, G., Bell, C., and Gray, L.: Enhanced signature of solar variability in Eurasian winter climate, Geophysical Research Letters, 37, 2010.

---

## Author Comment (AC3)

**Final Author Response**                                   **Van der Putten et al., cp-2021-69**

We take the opportunity offered to us to respond to the review reports of our paper.

Although our paper is rejected for further publication we have to address some of the comments by Referee 1 and 3 that we consider unjustified, as they undermine our scientific integrity without any basis. We believe that a significant number of comments are based on misunderstandings.

**Before going into detail we want to summarise our main points:**

- Referee 1 reproaches us of "*poor scientific practice*" as we are "*focusing on a single mechanism from the start without evaluating other possibilities*". We agree that in this paper we test the hypothesis of a Sun-climate link, based on chronologically well constrained data together with Reanalysis and Model data. How can testing a hypothesis be considered poor scientific practice? However, we agree that it could be stated more clearly that testing the Sun-climate link is the focus of this paper and that other possible forcing factors could be better discussed.

- Referee 1 and 3 accuse us of "*republishing old data*" and again, "*poor scientific practice*". Indeed, we have revisited a core from an exceptional site at 46°S in the southern Indian Ocean, for which part of the data were published in Van der Putten et al. (2008). However, at that time we could only speculate about solar forcing as our age-depth model was far from accurate enough. For our current paper we obtained a **high resolution chronology**, absolutely necessary for testing a Sun-climate link, together with **additional proxy-data** around the change of interest and supported by **Reanalysis and Model data**, to test our **hypothesis of a solar forced change in the Westerly wind belt**. Interestingly, in the reports of Referees 1 and 3 the words "Reanalysis" and "Model data" are not even mentioned.

- Referee 3 also states that our manuscript "*presents several conceptual and scientific flaws*" which in our opinion are based on misunderstandings by the referee on our proxy-data interpretation as we elucidate below. It seems that Referee 3 thinks that we present the results of a (atmospheric) dust record from a (ombrotrophic) peat bog which is not the case at all for our study as explained below.

Last but not least, we want to highlight that Referee 1 and 2 conclude that our data as well as our interpretations are sound. However, we do agree with Referee 2 that we have to *"provide additional details on our proxy data that will reinforce our interpretations"*. This would probably have avoided certain comments from Referee 3.

**Response to anonymous Referee 3**

We thank Referee 3 (R3) for the detailed and elaborate comments on all aspects of our study.

We refer to our reply to R1 for the aspects concerning the unfortunate accuses of "*poor scientific practice*" and "*re-publishing old data*". R3 also seems to ignore the new aspects of this work as the words "*high resolution dating*", "*Reanalysis*" and "*model data*" cannot be found in the review report.

R3 concluded that our paper *"presents several conceptual and scientific flaws"*, which we believe is based on the referee's misunderstandings on our data interpretation. We would like to emphasize

that R1 and R2 agree that our proxy-record is rigorous and that the interpretation of the data as increased westerly influence shortly after 2800 cal yr BP is sound.

However, we will address part of R3's comments below to hopefully clear up some misunderstandings.

**Site location and characteristics**

R3 states that *"The authors wish to study wind and precipitation within the westerly core belt. However, their site is clearly located to the east of the island, with mountain ranges to its west. This is far from ideal to study such processes. I therefore question how and why this site was selected, compared to potential sites to the west of the island with more directly influenced by westerly winds and precipitation, and how this location would ultimately affect the relevancy of the record."*

We do not agree that only sites from the western side of islands are relevant for SHW reconstructions. This is heavily overstated. Île de la Possession is a rather small island with its culminating point at about 900 m asl. The island is strongly influenced by the westerlies, on both the western and eastern coasts. The Morne Rouge volcano crater is a closed basin, located at the end of a huge U-shaped valley, and is perfectly orientated with the lowest point of the crater rim "facing" the west (fig. 1c and S4). We will add the meteorological data that are originating from the weather station at the base *Alfred Faure*, also located on the eastern side of the island (Fig. S4). The annual precipitation is high (2391 mm) as well as the mean wind velocity (9.6 m/s).
In some specific studies, it is necessary to obtain cores from the western side of the islands as the proxies used depend on past changes in the amount of sea-spray, used indirectly through diatom analysis (diatom inferred conductivity) as a proxy for wind strength (e.g. Saunders *et al.*, 2009; 2018). However this is not the case for our study.

**Proxy interpretation**

R3 refers to the following papers *Zaccone et al., 2013 QI; Zaccone et al. 2012 Plant and Soil; Leifeld et al., 2011 Plant and Soil, Sapkota 2006, PhD dissertation available online*, to state the following: *"Calculating a minerogenic flux is not as simple as the authors do from the LOI* and *So no, unfortunately, LOI cannot be used to reconstruct a minerogenic dust flux".* This comment is not relevant to our study since we are not reconstructing dust fluxes from ombrotrophic *Sphagnum*-dominated peat bogs, as is done in the proposed studies. Such ombrotrophic peat bogs do not occur on the Crozet archipelago (nor in the sub-Antarctic in general). A closed basin such as the small volcanic crater of the Morne Rouge, where our peat core was sampled, comes the closest to an "ombrotrophic" context and can be considered to relfect atmospheric conditions. It is depending, for its water/nutrient balance, on effective precipitation (precipitation *minus* evapotranspiration) and wind, bringing in minerogenics from the crater sides and/or from the vicinity of the volcano (see supplementary Fig. S4). In our peat record, the minerogenic content is mainly from a local source and we use it as a proxy for past wind and precipitation intensity on the island. The bulk of the minerogenic material consists of scoria (from the crater sides) and fine to coarse-grained sands, visible to the naked eye. So, in conclusion, we are not reconstructing atmospheric dust.

R3 states that "*the upper part of the sequence is a mire, in other words a minerotrophic peatland".* Yes, it is. However, the entire peat sequence analysed here is a mire, not just the upper part. The small volcanic crater is gradually filled by a mire that began to accumulate about 6000 cal yr BP (Fig. 1c in the manuscript). Shortly after 2800 cal yr BP a peat pond came into existence on the mire surface, caused by an intensification of westerly influence (higher precipitation and stronger winds). The pond persisted and is bordered by a mire, which has continually and gradually accumulated (deepening the pond/lake) until the present day. This is the mire we cored and analysed. In our peat sequence there

is thus a shift to wetter/windier conditions (plant macrofossils and increased flux of minerogenics), concomitant with the formation of the peat pond, explaining the mix of open-water, peat and moss-dwelling diatom species, starting shortly after 2800 cal yr BP.

R3 infers post-depositional processes linked to terrestrial diatoms: "*The authors actually give a clue that post-depositional processes exist as they find terrestrial diatoms in their peat profile, directly witnessing that indeed dissolution and mineral neoformation exist*". The Morne Rouge record is a peat record so terrestrial diatom species occur on the peat surface and are subsequently preserved in the accumulating peat deposits. No post-depositional processes are needed to explain this.

R3 also states "*Therefore, the only wind proxy presented here, that is derived from LOI, is invalid because 1/ different vegetation yield different ash content*". Our peat sequence is dominated by brown mosses from start to end thus this argument does not hold (Van der Putten *et al.*, 2008). We would like to stress that the peat-forming vegetation on sub-Antarctic islands differs from the peat-forming vegetation in climatically similar regions in e.g. southern South America and New Zealand; *2/ possible neoformation on root tissue (especially important in silicate rich areas – See Sapkota 2006)*, Sapkota 2006 is again a study on dust fluxes in bogs so we wonder on which root tissue, from which plant species, this neoformation should take place in our peat record; *3/ biogenic mineral neoformation (see my comment further below about diatoms)*", we do not understand why diatoms should be the product of neoformation. Diatoms occur on the mire surface and in the acrotelm and are subsequently preserved in the peat deposits (catotelm).

**Coring technique**

"*I am quite surprised by the coring technique which consist in pushing a 11cm diameter, 2.5-m long (and then 5-m long?) PVC tube in the peatland, which is against all the common "good practice" techniques used to retrieve undisturbed peat samples*".
We do not push the PVC tubes, but (carefully) hammer them into the peat in order to limit compaction as much as possible. Nonetheless, compaction does occur (which is a disadvantage of this coring technique). However, from a proxy-record point of view, compaction cannot "cause" a change in vegetation or diatom content. And considering our minerogenic proxies, we calculate fluxes (mg cm-2 yr-1), based on a very good age-model. In consequence, compaction does not alter the proxy-data. A huge advantage of this coring-method is that continuous cores of about 2,5 m long are recovered, which can easily be transported "closed", thus limiting oxidation, which offers the possibility of obtaining fresh (unoxidised) material in the lab when opening the cores, which can also be easily photographed and analysed on non-destructive core scanners.
"*Speaking of suction, I would like to know how the authors retrieved a 2.5-m and then a 5-m tube pushed in wet and rather sticky peat? Having cored peatlands quite a bit, I would say this is impossible without a motorized system, but nothing is described here. I would like to know how that was achieved to counterbalance the important suction*".
There is no suction as we retrieve the peat deposits around the PVC tube with a gouge, then rotate the tube to break the peat in the tube from the peat deposits below, before extracting the core. We do not need motorised systems and damage to the environment is no greater problem than when using a Russian corer, especially if the same volume of material is sampled, as is done with one PVC tube. Moreover, PVC tubes can be split in a work and archive part, allowing future research to be done on these peat sequences, an important consideration, taking into account the costs of logistics and people doing fieldwork on these remote islands. R3 suggests that the Russian corer should be used at all occasions. However, we believe that all coring techniques have their problems. In the case of minerogenic fens as the one of this study (but see also van der Putten *et al.*, 2015), a Russian corer does not work as it does not penetrate layers of coarser sediments embedded in the peat deposits and/or highly compacted peat layers. We also use a Russian corer, in the case of peat deposits that

consist of mainly organic matter (such as ombrotrophic raised bogs).

Last, but not least we want to reply to the following statement: *"There are records with equivalent depth and age resolution which are closer than DOME C. The same goes for the Northern hemispheric records".*
See previous comments for comparison with SH wide and NH records. Comparing/discussing our results in a broader SH context is not straightforward since, to our knowledge, except for Chambers *et al.*, 2007, no other record exists that has a sufficiently high resolution age-depth model. However, the Dome C ice-record is highly relevant to our study because we are comparing our island record (46° S) with the deuterium excess (d) data of the ice core. Deuterium excess is a proxy related to the moisture source area of the precipitation, with the source area for Dome C being north of 50° S in the Indian Ocean (lines 260-263 in the manuscript).

**References**

Chambers, F. M., Mauquoy, D., Brain, S. A., Blaauw, M., and Daniell, J. R. G.: Globally synchronous climate change 2800 years ago: Proxy data from peat in South America., Earth and Planetary Science Letters, 253, 439-444, 2007.

Saunders, K. M., Hodgson, D. A., and McMinn, A.: Quantitative relationships between benthic diatom assemblages and water chemistry in Macquarie Island lakes and their potential for reconstructing past environmental changes, Antarctic Science, 21, 35-49, 2009.

Saunders, K. M., Roberts, S. J., Perren, B., Butz, C., Sime, L., Davies, S., Van Nieuwenhuyze, W., Grosjean, M., and Hodgson, D. A.: Holocene dynamics of the Southern Hemisphere westerly winds and possible links to CO2 outgassing, Nature Geoscience, 11, 650-655, 2018.

Van der Putten, N., Hébrard, J.-P., Verbruggen, C., Van de Vijver, B., Disnar, J.-R., Spassov, S., Keravis, D., de Beaulieu, J.-L., De Dapper, M., Hus, J., Thouveny, N., and Frenot, Y.: An integrated palaeoenvironmental investigation of a 6200 year old peat sequence from Île de la Possession, Îles Crozet, sub-Antarctica., Palaeogeography, Palaeoclimatology, Palaeoecology, 270, 179-195, 2008.

Van der Putten, N., Verbruggen, C., Björck, S., Michel, E., Disnar, J.-R., Chapron, E., Moine, B. N., and de Beaulieu, J.-L.: The Last Termination in the South Indian Ocean: A unique terrestrial record from Kerguelen Islands (49°S) situated within the Southern Hemisphere westerly belt, Quaternary Science Reviews, 122, 142-157, 2015.